# *Opuntia* genus in Human Health: A Comprehensive Summary on Its Pharmacological, Therapeutic and Preventive Properties. Part 1

Eduardo Madrigal-Santillán [1,*], Jacqueline Portillo-Reyes [1], Eduardo Madrigal-Bujaidar [2], Manuel Sánchez-Gutiérrez [3], Paola E. Mercado-Gonzalez [3], Jeannett A. Izquierdo-Vega [3], Nancy Vargas-Mendoza [1], Isela Álvarez-González [2], Tomás Fregoso-Aguilar [2], Luis Delgado-Olivares [3], Ángel Morales-González [4], Liliana Anguiano-Robledo [1] and José A. Morales-González [1,*]

1   Escuela Superior de Medicina, Instituto Politécnico Nacional, "Unidad Casco de Santo Tomas", Ciudad de Mexico 11340, Mexico; jacke_star230990@hotmail.com (J.P.-R.); nvargas_mendoza@hotmail.com (N.V.-M.); languianorobledo@gmail.com (L.A.-R.)
2   Escuela Nacional de Ciencias Biológicas, Instituto Politécnico Nacional, "Unidad Profesional A. López Mateos", Ciudad de Mexico 07738, Mexico; edumadrigal.bujaidar@gmail.com (E.M.-B.); isela.alvarez@gmail.com (I.Á.-G.); tfregoso@ipn.mx (T.F.-A.)
3   Instituto de Ciencias de la Salud, Universidad Autónoma del Estado de Hidalgo, Ex-Hacienda de la Concepción, Tilcuautla, Pachuca de Soto 42080, Mexico; spmtz68@yahoo.com.mx (M.S.-G.); estephania.mercado@gmail.com (P.E.M.-G.); jizquierdovega@gmail.com (J.A.I.-V.); ldelgado@uaeh.edu.mx (L.D.-O.)
4   Escuela Superior de Cómputo, Instituto Politécnico Nacional, "Unidad Profesional A. López Mateos", Ciudad de Mexico 07738, Mexico; anmorales@ipn.mx
*   Correspondence: emadrigal@ipn.mx (E.M.-S.); jmorales101@yahoo.com.mx (J.A.M.G.); Tel.: +52-55-5729-6300 (ext. 62753) (E.M.-S.)

**Abstract:** *Opuntia* spp. is a diverse and widely distributed genus in Africa, Asia, Australia, and America. Mexico has the largest number of wild species; mainly *O. streptacantha*, *O. hyptiacantha*, *O. albicarpa*, *O. megacantha* and *O. ficus-indica*. The latter being the most cultivated and domesticated species. Throughout history, plants and their phytochemicals have played an important role in health care and *Opuntia* spp. has shown a high nutritional, medicinal, pharmaceutical, and economic impacts. Its main bioactive compounds include pigments (carotenoids, betalains and betacyanins), vitamins, flavonoids (isorhamnetin, kaempferol, quercetin) and phenolic compounds. Together, they favor the different plant parts and are considered phytochemically important and associated with control, progression and prevention of some chronic and infectious diseases. This first review (Part 1), compiles information from published research (in vitro, in vivo, and clinical studies) on its preventive effects against atherosclerotic cardiovascular diseases, diabetes and obesity, hepatoprotection, effects on human infertility and chemopreventive and/or antigenotoxic capacity. The aim is to provide scientific evidences of its beneficial properties and to encourage health professionals and researchers to expand studies on the pharmacological and therapeutic effects of *Opuntia* spp.

**Keywords:** *Opuntia* spp.; phytochemicals; biological activities; antioxidant capacity; chemopreventive effect

## 1. Introduction

The Traditional Medicine/Complementary and Alternative Medicine (TCAM) concept includes any practice, knowledge and belief in health that incorporates medicine based on plants, animals and/or minerals, spiritual therapies, manual techniques and exercises applied individually or in combination to improve human health. The World Health Organization (WHO) considers that TCAM have shown favorable factors that contribute to an increasing acceptance worldwide, such as easy access, diversity, relatively low cost and, most importantly, relatively low adverse toxic effects in comparison with allopathic

medicine where these effects are frequently attributed to synthetic drugs. For this reason, TCAM continues to be used by different populations to treat and/or prevent the onset and progression of chronic diseases including obesity, diabetes, hypertension, atherosclerosis, and cancer [1–3].

Throughout human history, plants and their phytochemicals have played an important role at improving human health care. *Opuntia* spp. species have specifically shown many beneficial properties and high biotechnological capacity. These plants classified as angiosperm dicotyledonous are the most abundant of the Cactaceae family and are importantly distributed in America, Africa, Asia, Australia, and in the central Mediterranean area. Due to their capacity to store water in one or more of their organs, they are considered "succulent plants" whose cultivation is ideal in arid areas since they are very efficient to generate biomass in water scarcity conditions [4–7].

Most opuntioid cacti have flat and edible stems called cladodes (CLD), paddles, nopales or stalks. Generally, young CLDs (also called nopalitos) are eaten as a vegetable in salads, while their fruits [called cactus pear fruits (tunas) or prickly pear fruits (PPFs)] are widely eaten as fresh seasonal fruit. Prickly pears are oval berries with lots of seeds throughout all the pulp and a semi-hard bark that contains thorns. They are grouped in different colors (red, purple, orange/yellow, and white). Generally, the fruit with white flesh and green skin is the most consumed as food [4–7]. Some evidences indicate that *Opuntia* plants have been consumed by humans for more than 8000 years and due to their easy adaptation and spread in different types of soil, their domestication process in man-made environments has increased favoring the constant collection of CLD and PPFs [7–10].

## 2. *Opuntia* genus in Mexico

*Opuntia* spp. is a diverse and widely distributed genus in the American Continent. However, Mexico has the largest number of wild species. The most representaive are *O. streptacantha* (OS), *O. hyptiacantha* (OH), *O. albicarpa* (OA), *O. megacantha* (OM) and *O. ficus-indica*. The latter is highly cultivated and domesticated species due to its nutritional, medicinal, pharmaceutical, and economic impacts. It is believed to be a secondary crop with fewer thorns derived from OM, (a native species from central Mexico) [4,7,8,10,11]. Currently, *O. ficus-indica* (OFI) has become as important a vegetable crop as corn and agave-*tequila*; its economic relevance is significantly increasing in our country and in other parts of the world, especially for improving health when nopal and prickly pear are included in a diet. Therefore, the OFI domestication process has favored changes in the texture, flavor, size, color, quantity and quality of the cladodes and their fruits [4,7,8]. Mexico and Italy are the main producing and consuming countries of the approximately 590,000 ha cultivated around the world. The Annual Mexican production can reach 350,000 tons; for this reason, our country represents approximately 90% of the total production worldwide. In addition, Mexico is the main producer of prickly pear, representing more than 45% of world production; however, only 1.5% of this production is exported [4,7,11,12].

## 3. Nutritional Value, Bioactive Compounds and Main Mechanisms of Action Involved

Various investigations where different extraction methods were used have documented the nutritional value of *Opuntia* spp. Most of these studies coincide in the differences among the phytochemical composition of their plant parts (fruits, roots, cladodes, flowers, seeds and stems) and the wild and domesticated species. These can be attributed to environmental conditions (climate, humidity), the type of soil that prevails in the cultivation sites, the age of maturity of the cladodes, and the harvest season [5–7]. In general, opuntioid cacti contain a large amount of water (80 and 95%), carbohydrates (3–7%), proteins (0.5–1%), soluble fiber (1–2%), fatty acids (palmitic, stearic, oleic, vaccenic and linoleic) and minerals [Potassium (K), calcium (Ca), phosphorus (P), magnesium (Mg), chrome (Cr) and sodium (Na)]. They also have viscous and/or mucilaginous materials [made up of D-glucose, D-galactose, L-arabinose, D-xylose and polymers such as β-D-galacturonic acid linked to (1–4)

and L-rhamnose residues linked with R (1–2)] whose function is to absorb and regulate the amount of cellular water in dry seasons [5–7,13]. Among the main bioactive compounds of prickly pear highlight the pigments (carotenoids, betalains, betaxanthins and betacyanins), vitamins (B1, B6, E, A, and C), flavonoids (isorhamnetin, kaempferol, quercetin, nicotiflorine, dihydroquercetin, penduletin, lutein). Rutin, aromadendrine, myricetin vitexin, flavonones and flavanonols) and phenolic compounds (ferulic acid, feruloyl-sucrose and synapoyl-diglycoside) [5–7,13–16].

Specifically, the CLDs and prickly pears of OFI have shown several kinds of bioactive compounds, among which flavonoids (such as quercetin, kaempferol, isorhamnetin), essential amino acids [Glutamine (Glu), arginine (Arg), leucine (Leu), isoleucine (Ile), lysine (Lys), valine (Val), and phenylalanine (Phe)], vitamins (B1, B6, E, A, and C), minerals (mainly K and Ca), and betalains [such as betaxanthins (betanin and indicaxanthin) and betacyanins (betanidin, isobetanin, isobetanidine, and neobetanin) (Table 1) [5–7,13–16].

**Table 1.** Main bioactive compounds in different anatomical parts of *O. ficus-indica.*

| Chemical Group | Bioactive Compound |
|---|---|
| Cladodes (CLD) | |
| Flavonoids | Quercetin (2.0–40 mg/100 g), isoquercetin (2.29–39.67 mg/100 g), isorhamnetin-3-O-glucoside (4.59–32.21 mg/100 g), kaempferol (2.2 g/kg), nicotiflorin (2.89–146.5 mg/100 g), rutin (2.36–26.17 mg/100 g) |
| Phenoliccompounds | Gallic acid (0.6–2 mg/100 g), coumaric (14–16 mg/100 g), 3,4-dihydroxybenzoic (0.06–5.0 mg/100 g), 4-hydroxybenzoic, ferulic acid (0.5–34 mg/100 g) |
| Amino acids | Glu (36 g/100 g), Arg (5.0 g/100 g), Leu (2.7 g/100 g), Ile (3.97 g/100 g), Lys (5.2 g/100 g), Val (7.7 g/100 g), Phe (3.5 g/100 g), Glu (36.1 g/100 g |
| Minerals | K and Ca (mainly calcium oxalate crystals). Amounts ranging from 230 to 5500 mg/100 g |
| Vitamins | E (2182 mg/100 g), A (7–10 mg/100 g), C (7–22 mg/100 g), B1 (0.14 mg/100 g), B2 (0.60 mg/100 g), B3 (0.46 mg/100 g) |
| Prickly pear fruits (tunas) | |
| Flavonoids | Kaempferol(53.2 mg/100 g), quercetin(90 µg/g), isorhamnetin(49.4 µg/g) |
| Amino acids | Lys (0.63 g/100 g), Met (2.0 g/100 g), Glu (12.5 g/100 g), Taurine (15.7 g/100 g) |
| Minerals | K (161 mg/100 g), Ca (27 mg/100 g), Mg (27 mg/100 g) |
| Vitamins | E (527 mg/100 g), A (5 mg/100 g), C (34–40 mg/100 g) |
| Organic acids | Maleic, malonic, succinic, tartaric, and oxalic. Total average of 0.36 to 8.50 mg/g |
| Betalains | Betaxanthins (25.4 mg indicaxanthins/100 g), Betacyanins (15.2 mg betanin/100 g), Betalains (40.6 mg/100 g) |
| Seeds | |
| Phenolic compounds | Ferulic acid (3–17 mg/100 g), sinapoyl-diglucoside (13–22 mg/100 g), synapoyl-glucose, feruloyl-sucrose (8–17 mg/100 g). Total average of compounds 48–89 mg/100 g |
| Minerals | Mainly K and P (160–150 mg/100 g). Lower proportions of Mg (70 mg/100 g), Na (66 mg/100 g) and Ca (16 mg/100 g) |
| Sterols | β-sitosterol (67.0–21.0 g/kg) and campesterol (1.6–8.7 g/kg) |
| Fatty acids | Palmitic acid (9–20 g/100 g), oleic acid (16–18 g/100 g), linoleic acid (53–70 g/100 g) |
| Pulp and peel | |
| Flavonoids | Quercetin (4–9 mg/100 g), isorhamnetin (3–90 mg/100 g), kaempferol (0.2–0.8 mg/100 g), luteolin (0.8–1.0 mg/100 g), isorhamnetin glycosides (50–60 mg/100 g) |
| Phenolic compounds | Ferulic acid, sinapoyl-diglucoside, feruloyl-sucrose isomer. Total average of compounds 218.8 mg/100 g |
| Minerals | More K (161 mg/100 g) than Ca and Mg |
| Sterols | β-sitosterol (67.0–21.0 g/kg) and campesterol (1.6–8.7 g/kg) |
| Fatty acids | Palmitic acid (34 g/100 g), oleic acid (10.8 g/100 g), linoleic acid (37 g/100 g), linolenic acid (12.6 g/100 g) |

**Table 1.** *Cont.*

| Chemical Group | Bioactive Compound |
| --- | --- |
| | Flowers |
| Flavonoids | Kaempferol (300–400 mg/100 g), quercetin (400–700 mg/100 g), isorhamnetin glycosides [isorhamnetin 3-O-robinobioside (4269 mg/100 g), isorhamnetin 3-O-galactoside (979 mg/100 g), isorhamnetin 3-O-glucoside (724 mg/100 g)] |
| Organic acids | Mainly gallic acid (1600–4900 mg/100 g) |

Table modified from El-Mostafa et al. (2014) [6]. Average and most significant data obtained from Angulo-Bejarano et al. (2014) and El-Mostafa et al. (2014) [5,6]. Potassium (K), Phosphorus (P), Magnesium (Mg), Sodium (Na), Calcium (Ca), methionine (Met), glutamine (Glu), arginine (Arg), leucine (Leu), isoleucine (Ile), lysine (Lys), valine (Val), and phenylalanine (Phe).

Various studies have shown the action of phytochemicals as substrates to activate different biochemical reactions that provide important health benefits. For that reason, they could be included in the definition of nutraceutical: "Any non-toxic food extract supplement that has been scientifically proven to be beneficial to health both intreating and preventing diseases" [5,17].

In this context, opuntioid cacti reveal different mechanisms of action that can be interrelated and favor their biological effects. In general, they are organized in 7 groups: (I) Inhibition of the absorption of substances, favoring the absorption of protective agents and/or modification of the intestinal flora (action of soluble fiber and ascorbic acid), (II) Scavenging of reactive oxygen species and/or protection of DNA nucleophilic sites (antioxidant action), (III) Anti-inflammatory activity, (IV) Modification of transmembrane transport (effect of short-chain fatty acids and calcium in the diet), (V) Modulation of xenobiotic metabolising enzymes, inhibition of mutagen agents activation and induction of detoxification pathways (flavonoids, polyphenols and índoles), (VI) Enhancement of apoptosis (action of some flavonoids), and (VII) Maintenance of genomic stability (effect of some vitamins, minerals and polyphenols) [3,5–7,14–17].

Due to the bioactive compounds of *Opuntia* spp. the different plant parts can be associated to control, progression, and prevention of chronic and infectious diseases. However, it is relevant to comment that the Cactaceae family contains approximately 130 genera and 1500 species, which favors a wide genetic diversity that in conjunction with environmental conditions (climate, humidity), soil type, age of maturity of the cladodes and the harvest season generates differences in the phytochemical composition between wild and domesticated specie, inducing changes in their nutritional values and/or functional properties. This first review (Part 1), focuses on information from published research (in vitro, in vivo and clinical studies) on its action in atherosclerotic cardiovascular diseases, diabetes and obesity, hepatoprotection, effects on human infertility and chemopreventive and/or antigenotoxic capacity; which will be discussed below.

**4. Pharmacological, Therapeutic and Preventive Properties**

*4.1. Effects on Atherosclerotic Cardiovascular Diseases*

The term atherosclerotic cardiovascular diseases (ASCVD) encompasses conditions that affect the blood vessels and the heart as a consequence of the thickening and hardening of medium and large-caliber arteries. ASCVD involves diseases of the cardiovascular system that share similar characteristics regarding their cause, pathophysiology, prognosis, and treatment. They are one of the main causes of mortality in the world (including Mexico) and unfortunately, their incidence is increasing [18].

In general, atherosclerosis is an inflammatory and chronic process characterized by the progressive occlusion of arteries where there is retention, oxidation, and modification of lipids in the form of fatty stretch marks, whose development can generate endothelial dysfunction, inflammation, and thrombosis. When serum concentrations of LDL-cholesterol (LDL-Cho) rises significantly, it penetrates the arterial walls to accumulate among the cells where reactive oxygen species (ROS) are induced and produce oxidation of LDL

(Low-density lipoprotein) that generates the release of pro-inflammatory cells (such as monocytes and neutrophils).

Specifically, monocytes become macrophages, which promote the progression of the lesion by stimulating an inflammatory cascade. Various situations that cause endothelial damage have been identified, such as hypercholesterolemia, hypertension, diabetes, obesity, and smoking [7,18,19], which together with genetic predisposition are considered traditional risk factors; Several of these factors are related to changes in lifestyle, which has contributed to generating certain strategies (especially those focused on smoking and poor nutritional habits) that reduce the possibility of cardiovascular risk. In this context, both CLD and PPFs from *Opuntia* spp. have shown antiatherogenic, antihyperlipidemic, and antihypercholesterolemic properties due to their soluble fiber content, flavonoids, phenolic compounds and fatty acids. In the case of fiber, composed of substances (such as cellulose, hemicelluloses, pectin, lignin and gums) resistant to digestive enzymes, the hypolipidemic effect is attributed to them due to their binding to dietary fat which promotes its excretion by fecal route with the consequent reduction of body fat [7,14,20,21].

On the other hand, some flavonoids, phenolic compounds and fatty acids, are considered phytochemicals with the same activity and anti-inflammatory effects. Such properties are supported by its antioxidant capacity. Specifically, quercetin 3-methyl ether, obtained from cladode extracts of OFI, has shown potential to lower cholesterol (Cho); while omega-6 linoleic acid from cactus seed oil is considered a precursor of arachidonic acid with a hypocholesterolemic effect [6,7]. Also, betalains [such as betanin and indicaxanthin (Ind)] from PPFs have evidenced to protect the vascular endothelium from inflammation and cytokine-induced oxidative alteration [such as tumor necrosis factor alpha (TNF-$\alpha$)] by inhibiting intercellular adhesion molecule-1 (ICAM-1) [22].

Table 2 shows the most relevant studies of the hypolipidemic, hypocholesterolemic and antiatherogenic properties of *Opuntia* spp. In summary, from 1996 to date, 4 out of 17 have been in vitro studies; 7, using laboratory animals (mainly rodents); 5, developed with patients (clinical studies) and only one where a systematic review was made. Garcia-Diez et al. (1996) were the first researchers to explore a diet supplemented with pectin (extracted from nopal) in the metabolism of Cho and bile acids of Wistar rats. After four weeks, they obtained lower concentrations of Cho in serum and liver; as well as a higher activity of the regulatory enzymes of Cho [3-hydroxy-3-methylglutaryl-CoA reductase (HMG-CoA reductase) and cholesterol 7 alpha-hydroxylase (Cho7AH) [23]. Possibly, these results prompted other researchers to analyze the intake of PPFs (in pulp form) in patients suffering from isolated heterozygous familial hypercholesterolaemia [24] and primary hypercholesterolaemia [25]; as well as to extract a glycoprotein (GOFI) from the nopal to be orally administered to mice for two weeks and to show that in both cases the plasma levels of triglycerides (TG), tCho and LDL were reduced [26]. In 2012, the lethal dose 50 (LD$_{50}$) of a methanolic extract of *O. joconostle* (OJ) seeds was found to be greater than 5000 mg/kg and that mice fed with a hypercholesterolemic diet along with this oral extract could reduce the concentrations of TG, tCho and LDL-Cho [27]. A similar phenomenon took place when evaluating a food supplement (*NeOpuntia*) obtained from dehydrated leaves of OFI on blood lipid parameters of 59 women after 6 weeks of treatment [28].

Regarding the antiatherogenic potential, during 2015 this property was tested by using different powdered cladodes of *Opuntia* (OS, OH, OA, OM, OFI) against the oxidation of LDL caused by vascular endothelial cells and the toxicity of 4- hydroxynonenal under normal conditions (Apc +/+) and in immortalized preneoplastic epithelial colon cells (Apc min/+). The conclusion was that all powders significantly inhibited the oxidation induced by incubation with murine endothelial cells and the formation of foam cells of murine macrophages RAW 264.7 [29].

**Table 2.** Studies testing for hypolipidemic and antiatherogenic effects of *Opuntia* spp.

| Type of Study | Objective and Characteristics | Results and Conclusion | Ref. |
|---|---|---|---|
| | Hypolipidemic and hypocholesterolemic effect | | |
| In vivo | The objective was to evaluate the effect of pectin (extracted from CLD) in the metabolism of Cho and bile acids in male Wistar rats. The animals were fed during 4 weeks with a pectin supplemented diet (7 g/100 g) where Cho regulatory enzymes (HMG-CoA reductase and Cho7AH), concentrations of circulating and hepatic lipids and the excretion of fecal bile acids were measured. | After this period, lower concentrations of Cho were found in serum and liver, as well as a significant excretion of fecal bile acids and a greater activity of Cho7AH and HMG-CoA reductase. It was concluded that pectin favors the decrease of hepatic cholesterol and its serum concentrations. | [23] |
| Clinical study | Considering that the PPFs is a traditional food of the American indigenous population, the effect of its pulp was evaluated in 15 patients of both genders who suffered from isolated heterozygous familial hypercholesterolemia (FH). | After a daily consumption of 250 g of pulp for four weeks, tCho and LDL-Cho had diminished in all patients, being more significant in men. Likewise, when analyzing the oxidative damage using the biomarker 8-epi-PGF (2 alpha), a decrease in plasma, serum, and urine was evidenced. The results suggest that the consumption of PPFs can reduce oxidative lesions and benefit the cardiovascular system. | [24] |
| Clinical study | In this pilot study the aim was to evaluate the effects of PPFs on lipid metabolism. A group of 24 men with FH (without diabetes and obesity) consumed its pulp (250 g/day) for eight weeks. | After this period, a significant reduction of tCho (12%), LDL-Cho (15%), TG (12%) and apolipoprotein B (9%) was observed. The conclusion was that PPFs may show a hypocholesterolemic action due to the content of soluble fiber (such as pectin). | [25] |
| In vivo | A glycoprotein (GOFI) was isolated from *O. ficus-indica var. Saboten* to determine its ability to reduce the plasma lipid level through scavenging of intracellular radicals in Triton WR-1339-induced mice. GOFI was orally administered to the animals (50 mg/kg) for two weeks. | The results showed that GOFI reduced the plasma levels of triglycerides (TG), total concentration of cholesterol (tCho) and Low-density lipoproteins (LDL) induced by Triton WR-1339. In addition, a decrease in the level of thiobarbituric acid-reactive substances (TBARS) and an increase in the enzymatic activity of superoxide dismutase (SOD), catalase (CAT) and glutathione peroxidase (GPx) were observed. The hypolipidemic effect is probably related to the antioxidant capacity of GOFI. | [26] |
| In vivo | The purpose of this study was to evaluate the hypolipidemic effect of a methanol extract (MeOH) from *O. joconostle* (OJ) seeds in mice fed with a hypercholesterolemic diet. | It was initially found that the oral lethal dose 50 ($LD_{50}$) was greater than 5000 mg/kg. The supplementation of the extract (doses of 1, 2 and 5 g/kg) significantly decreased the concentrations of TG, tCho and LDL-cholesterol (LDL-Cho). The hypolipidemic effect of MeOH was probably due to the phenolic composition of the seeds and the dose administered. | [27] |
| Clinical study | Considering that NeOpuntia is a nutritional supplement obtained from dehydrated leaves of OFI, the purpose of this monocentric study, randomized, and placebo-controlled was to analyze its effect on blood lipid parameters of 59 women. During 6 weeks, the individuals consumed balanced diets with controlled lipid intakes plus NeOpuntia capsules (1.6 g dose per meal) and TG, LDL-Cho, and HDL-Cho (High-density lipoprotein cholesterol) levels were measured. | Most of the women showed an increase in HDL-Cho levels and a decrease in blood levels of TG and LDL-Cho. The results suggest that NeOpuntia may reduce cardiovascular risks. | [28] |

**Table 2.** *Cont.*

| Type of Study | Objective and Characteristics | Results and Conclusion | Ref. |
|---|---|---|---|
| Clinical study | After subjecting 10 patients with FH to a dietary treatment with PPFs for 6 weeks, the hypolipidemic potential of this fruit was evaluated by the uptake of autologous (123) I-radiolabeled LDL. | The results showed a relevant increase in the hepatic uptake of LDL and consequently, lower levels of tCho and LDL-Cho in the circulating blood. These findings suggest that the beneficial effect is related to a positive regulation of the receptor [(123) I-LDL]. | [30] |
| In vivo | Stem intake of *O. humifusa* (OHF) was examined on the regulation of lipid concentrations in Sprague-Dawley rats with streptozotocin (STZ) injection-induced diabetes mellitus (DM). The animals were treated orally with two doses of OHF (150 and 250 mg/kg per day) for seven weeks. | Both treatments favored a level lowering of TG, tCho and LDL-Cho. Furthermore, the alanine aminotransferase (ALT) and aspartate aminotransferase (AST) concentrations were significantly reduced compared to the DM control group. These results suggest that OHF is potentially hypolipidemic. | [31] |
| In vivo | The capacity of ODP-Ia (main component of *O. dillenii Haw* (OdHw) polysaccharides) on lipid concentration in hyperlipidemic rats induced by high-fat emulsion was analyzed. | After the oral administration of ODP-Ia, serum lipid levels and liver concentrations of tCho and TG significantly decreased. The same treatment increased the activity of cholesterol acyltransferase and SOD (serum and hepatic) and inhibited the action of HMG-CoA reductase and the content of malondialdehyde (hepatic and serum). In addition, by means of a histopathological analysis, the inhibition of the infiltration of inflammatory cells was observed. Together, these results suggest that ODP-Ia is a natural product that can be used in the treatment of hyperlipidemic diseases and that their mechanisms of action are related to the antioxidant potential and the modulation of the enzymes involved in the metabolism of Cho. | [32] |
| In vitro | In this test, the effect of piscidic acid and some derivatives of isorhamnetin (Isorhamnetin glucosyl-rhamnosyl-rhamnoside, isorhamnetin-glucosyl-rhamnosyl-pentoside, isorhamnetin-3-O-glucosyl-pentoside, Isorhamnetin-3-O-rutinoside) was evaluated on the absorption of Cho in a monolayer of Caco-2 cells. | The results indicated an approximate 38% reduction in Cho permeation, while for phenolic compounds it was 6% (isorhamnetin) and 9% (piscidic acid). It was also observed that the mixture of both phytochemicals showed an IC50 of 20.3 μg/mL (inhibition of the HMG-CoA enzyme), while for the piscidic acid it was 149.6 μg/mL. This value was slightly exceeded by isorhamnetin derivatives. The data suggest considering OFI as a promising plant for the development of new pharmaceuticals with hypocholesterolemic potential because its bioactive compounds could bind to the active site of the HMG-CoA enzyme. | [33] |
| Systematic review | Despite the concise benefits of *Opuntia* spp. in ASCVD, there is still some confusion about the lipid-lowering effect between its CLD and PPFs. Due to that confusion a systematic review of the characteristic was carried out (from February to September 2019) in the main electronic databases, considering both plant parts and using keywords such as tCho, LDL-Cho, HDL-Cho and TG. Eleven articles (6 from PPFs, 4 from CLD and 1 from commercial products) met the established criteria. | In summary, the consumption of PPFs is associated with significant reductions in tCho, LDL-Cho and TG; while in CLD the lipid-lowering effect is less and there is a datum on a significant increase in HDL-Cho. Possibly, the discrepancies in this effect are caused by the different chemical compositions between CLD and PPFs. Therefore, it would be more feasible to identify the components of *Opuntia* spp. with greater precision in future studies. | [34] |

**Table 2.** *Cont.*

| Type of Study | Objective and Characteristics | Results and Conclusion | Ref. |
|---|---|---|---|
| In vivo | The content of the total phenolic content and the antioxidant and antihyperlipidemic activities of the seed oil of *O. dillenii Haw* (OdHw) were evaluated. Using the 2,2-diphenyl-1-picrylhydrazyl (DPPH) scavenging assay and the Folin-Ciocalteu method, the antioxidant activity and phenolic content were tested. The other property was evaluated in albino mice fed a high-fat diet plus OdHw (2 mL/kg). | The oil showed a high phenolic content and DPPH scavenging activity. It also presented a significant antihyperlipidemic effect by improving the lipid profile of the animals; which suggests that this property is related to the antioxidant activity and the phenolic content of the plant's seeds. | [35] |
| | | Antiatherogenic effect | |
| In vitro | As is known, atherosclerosis is a chronic process where macrophages stimulate inflammatory cascades that promote endothelial dysfunction and allow the constitutive form of ICAM-1 to be expressed. Therefore, the ability of two betalains (betanin and Ind) from PPFs to protect the endothelium from cytokine-induced oxidative alteration by inhibiting ICAM-1 was evaluated. Human umbilical vein endothelial cells (HUVECs) were stimulated with TNF-$\alpha$ and flow cytometry measurements were subsequently performed by incubation with anti-human-ICAM-1. | The results showed that both pigments were able to slightly inhibit ICAM-1 expression up to a micromolar concentration. The antioxidant evaluation of these phytochemicals opens the possibility to develop pharmacological studies that are related to other pathologies characterized by endothelial dysfunction such as atherothrombosis, low limb ischemia, and stroke. | [22] |
| In vitro | The purpose of the study was to investigate the protection of different *Opuntia* CLDs (OS, OH, OA, OM, OFI) in powder form against LDL oxidation caused by vascular endothelial cells and the toxicity of 4-hydroxynonenal under normal conditions (Apc +/+) and in preneoplastic immortalized epithelial colon cells (Apc min/+). | All powders showed a significant inhibition of the oxidation induced by incubation with murine endothelial cells and the foam cell formation of RAW 264.7 murine macrophages. Furthermore, they reduced murine endothelial cell cytotoxicity and colon cancer development in the in vitro model. The conclusion was that the therapeutic potential of cladodes is related to their antioxidant capacity and their content of phenolic acid and flavonoids. | [29] |
| In vitro | Since macrophage apoptosis induced by 7-ketocholesterol (7-KC) is a key event in the development of human atheromas, the study of the effect of Ind on 7-KC-induced apoptosis of human monocyte/macrophage THP-1 cells was considered. The proapoptotic potential of 7-KC was evaluated by cell cycle arrest, phosphatidylserine exposure in the plasma membrane, variation of nuclear morphology, and activation of the antagonist Bcl-2 (B-cell lymphoma 2) of cell death. | During the first 24 h, elevated ROS levels were observed, preceding the overexpression of NADPH oxidase-4 (NOX-4) and the elevation of cytosolic $Ca^{2+}$; confirming the 7-KC-dependent activation of the redox-sensitive NF-$\kappa$B; while the co-incubation of Ind (2.5 $\mu$m) prevented such pro-apoptotic events. This pigment of PPFs might protect against atherogenic toxicity of 7-KC by inhibiting overexpression of NOX-4, inhibiting the activation of Nuclear Factor Kappa-Light-Chain-Enhancer of Activated B Cells (NF-$\kappa$B), the maintenance of cellular redox balance and $Ca^{2+}$ homeostasis. | [36] |
| In vivo | The purpose of the study was to determine whether CLDs of OS and OFI could prevent the development of atherosclerosis in ApoE(-)KO mice. Likewise, using both *Opuntia* species, the concentration of ROS, the kinetics of LDL oxidation by murine CRL2181 endothelial cells, and the capacity of the inflammatory process to induce the adhesion of monocytes in the activated endothelium and the formation of foam cells were determined. | The evidence showed that OS and OFI had significantly reduced the extracellular generation of superoxide anion, the oxidation of LDL and its subsequent signaling cascade (including the expression of ICAM-1 and NF$\kappa$B). A reduction in atherosclerotic lesions and 4-hy-droxynonenal adducts in the vascular wall of mice was also observed. Therefore, it is suggested that both *Opuntia* species (wild and domesticated) show antioxidant, anti-inflammatory, and anti-atherogenic properties. | [37] |

| Type of Study | Objective and Characteristics | Results and Conclusion | Ref. |
|---|---|---|---|
| Clinical study | Considering that the antioxidant properties of OFI have been associated with a reduction in body fat, the effects of a dietary supplementation of a 3% OFI extract (500 g per week) was analyzed in 49 individuals (13 men and 36 women). The complete profile of the LDL subclass was evaluated by gel electrophoresis for one month. | The study showed in a percentage increase of LDL-1 and a concomitant reduction in LDL-2; which suggest that OFI extract may have beneficial effects on LDL particle size making them less atherogenic. | [38] |

### 4.2. Effects on Diabetes and Obesity

The metabolic syndrome is the set of carbohydrate and lipid metabolic abnormalities that describes the increasing incidence of type 2 diabetes mellitus (DM2) associated with abdominal obesity (AO), insulin resistance, and cardiovascular disease [14,39]. DM2 is a multifactorial disease that includes genetic determinants of individual susceptibility and environmental factors of lifestyle. It is considered a serious public health problem, being among the leading causes of death worldwide.

The high social costs and implications for hospital systems are some of the relevant consequences. The worldwide increase of the metabolic syndrome is estimated to 360 million people by 2030. It is characterized by a decrease in glucose uptake stimulated by insulin (insulin resistance) and once the disease is established and there is poor control in the patients, macrovascular complications may develop (including atherosclerosis) as well as microvascular abnormalities (such as retinopathy, nephropathy, and neuropathy). Furthermore, in the long term, grave problems may result in the kidney and the heart [5,7,14,39].

On the other hand, obesity (also considered a public health problem) has significantly increased as a result of rapid urbanization, growing technology, altered eating habits, and decreasing physical activity. Specifically, Mexico faces a challenging situation due to its high incidence. The statistical data indicate that in adults it occurs in 33% (higher prevalence in women) and approximately 15% in children [7].

There is a close relationship between AO and insulin resistance. This is due to the fact that AO implies an increase in fat at the visceral level (mainly in the liver, muscles and pancreas), which induces the formation of adipokines in the fat tissue and favors pro-inflammatory and prothrombotic states, which contribute to the development of insulin resistance, hyperinsulinemia and endothelial dysfunction (cardiovascular disease) [5,39].

Adiponectin, in particular, decreased; which is a situation associated with an increase in TG, a decrease in HDL, an elevation of apoliprotein B, and the presence of dense LDL particles, contributing to the atherothrombotic state that represents the inflammatory profile of visceral adiposity [39]. It is also known that oxidative stress (OXs) is related to insulin resistance and obesity and that a high concentration of ROS can induce and/or favor the development of both diseases.

An interesting observation is that low levels of adiponectin are usually the result of this high concentration; also, the production of ROS in adipocytes is associated with insulin resistance and alterations in serum levels of adiponectin with the consequent inflammatory response [5,40].

The herbal treatments and traditional plant-based medicines are increasingly popular due to their low-cost with apparently fewer side effects for treating DM2 and AO. In general, four possible mechanisms of action have been directed at *Opuntia* spp.

The first is related to its content of fiber, pectin and mucilage that slow down the speed of digestion and/or intestinal absorption of glucose and fatty acids [5,7,25,34,41,42]. The second focuses on improving the postprandial glucose response and stimulating insulin secretion through a direct action on pancreatic β cells after a dietary intake that includes nopal [43–45].

Another hypothesis lies on its antioxidant properties. As it is mentioned above, OXs plays a fundamental role in the development of atherosclerosis and cardiovascular diseases, the main complications of DM2 and AO [40]. A relevant data on this mechanism is its content of polysaccharides (arabinose, xylose, fructose, glucose, galacturonic acid and rhamnose) that have demonstrated anti-inflammatory activity and ability to isolate ROS. This supports the fact that these polysaccharides can reduce hepatic lipoperoxidation, maintaining the tissue function and improving the target cells response to insulin [46,47]. The final proposal is that Cr (III), present in the CLD of nopal and extracts of the pulp of PPFs, is an important element in mammals to maintain the balance of carbohydrates and lipids. Some studies have validated this property, finding a positive effect of Cr (III) on insulin signaling and/or function; helping to improve its systemic sensitization and reduce plasma glucose under fasting conditions [48–50].

After the studies related to cardiovascular diseases, the greatest scientific interest in *Opuntia* spp. has focused on its ability to treat DM2 and AO. Approximately 47 scientific articles have been published since the 1980s; 10 of which belong to clinical investigations, 4 are in vitro studies, 24 in vivo tests (using rodents, rabbits and pigs), and 9 of them are systemic reviews.

Initially, the research of Ibañez and Meckes (1983) stands out, who evaluated the hypoglycemic effect of a semi-purified fraction of OS in rabbits and observed that the powder obtained would produce an effect similar to the traditional extract of OS stems [51]. Years later, with the same animal model, another study confirmed that OS decreased the area under the glucose tolerance curve and the hyperglycemic peak [52].

Likewise, they observed that the red juice obtained from *O. dillenii Haw* (OdHw) increased plasma insulin levels in normoglycemic and alloxane (Allox) -induced diabetic rabbits [53]. Another frequently used experimental model is the administration of strepto-zocin (STZ) to induce diabetes in rodents. A dose of 1.0 mg/kg/day of a purified extract of *O. fuliginosa* (Of) decreased blood glucose levels and glycosylated hemoglobin (HbA1c) in diabetic rats [54].

Not only the species Of has demonstrated this property, since the juice of OdHw, a rich source of fiber, minerals and vitamins, has also reduced glucose levels [55]. On the other hand, four extracts of *O. Milpa Alta* [Aqueous, petroleum ether, ethyl acetate (EtOAc), and butanol (BuOH)] were tested in STZ-induced diabetic mice and they also managed to lower glucose levels [56].

Hahm et al. (2011) studied the intake of three doses of OHF (150, 250 and 500 mg/kg/day) for 7 weeks on the regulation of blood glucose in diabetic rats; their conclusion was that all doses reduced this blood parameter to values comparable to the DM control group. In particular, the group treated with 500 mg/kg showed a considerable increase in the relative volume of β pancreatic cells [31].

This final result was also confirmed by Yoon et al. (2011) who, when administering an OFI extract for 4 weeks to db/db mice and performing a histopathological analysis, observed that the morphology of the pancreatic islets were significantly improved [57].

In studies related to AO, the absorption of fat in the diet decreased through natural treatments, such as Litramine IQP-G-002AS (fiber derived from OFI). The results of four randomized clinical studies suggest that it is effective in promoting fat excretion and weight loss when taken at a daily dose of 3 g for seven days [58].

Another relevant study showed that an OFI extract included in a high-fat diet and administered for 12 weeks to C57BL/6 mice, prevented the rise of body weight and blood levels of LDL-Cho, HDL-Cho, tCho. In addition, the extract stimulated insulin secretion produced by their pancreatic islets [59].

Given that the information is very extensive, only the most significant documents will be analyzed, so the summary of the rest of the investigations, including in vitro and in vivo trials, clinical studies, and systematic reviews are included in Table 3.

**Table 3.** Main studies of *Opuntia* spp. on its effects in diabetes and obesity.

| Type of Study | Objective and Characteristics | Results and Conclusion | Ref. |
|---|---|---|---|
| | Diabetes | | |
| Clinical study | This is a pilot study where the effect of PPFs consumption on glucose metabolism was observed in 24 non-diabetic and non-obese men with FH. | The results showed a decrease in blood glucose (11%), insulin (11%), uric acid (10%) and TG (12%); while their body weight and HDL-Cho remained unchanged. This hypoglycemic action is related to an improvement in insulin sensitivity, and possibly to the pectin contained in PPFs. | [25] |
| In vivo | This study was on the intake of three doses of OHF (150, 250 and 500 mg/kg/day) regarding the regulation of blood glucose and hypolipidemic responses in diabetic rats induced by STZ. | After 7 weeks of oral treatment, fasting TG and blood glucose levels were significantly lower compared to the DM control group. In addition, the group treated with 500 mg/kg showed an increase in the relative volume of pancreatic β cells. | [31] |
| In vivo | The total phenolic content and the antioxidant activity of the seed oil of OdHw were analyzed using the Folin-Ciocalteu method and DPPH-scavenging assay, respectively. Also, the preventive effect of OdHw against alloxane (Allox)–induced DM was evaluated in albino mice. | The results showed that it had a high phenolic content and significant DPPH purifying activity. Likewise, the loss of body weight and the mortality rate caused by Allox decreased and the blood sugar level was controlled; which suggests that these protective actions are related to its phenolic content. | [35] |
| Clinical study | The hypoglycemic effect of *O. streptacantha* (OS) was evaluated in 16 patients with non-insulin-dependent diabetes mellitus (NIDDM). | Their serum glucose and insulin levels were quantified at 0, 60, 120 and 180 min. The result was that both parameters significantly decreased in individuals who ingested 500 g of roasted nopal stems. It was suggested to extend the studies in order to clarify the mechanism of action of OS. | [41] |
| Systematic review | The aim of this review was to identify the effects of *Opuntia* spp. consumption on glucose and insulin in humans, taking into account components such as PPFs, CLD and combined products. | During the research with six electronic databases, twenty articles were obtained (4 with PPFs, 12 with CLD and 4 with other products) that demonstrated a relevant reduction in serum glucose and insulin. The conclusions were that studies that specifically use PPFs or CLD have a high risk of bias. Apparently, PPFs have no significant effects on these parameters; unlike cladodes, which are more promising for hypoglycemic effects. | [42] |
| In vivo | Considering that only few data exist on OFI stem and fruit preparation combinations, the purpose of this study was to investigate the effects of an aqueous extract of CLD and a patented fruit stem/skin blend (ratio: 75/25) on blood glucose and plasma insulin in normal rats. | The observations were that the aqueous extract lowers glucose in a range of 6 and 176 mg/kg; while the patented blend was at 6 mg/kg. In addition, the mixture increases plasma insulin levels. The results suggest that both extracts have hypoglycemic activity, but the potential of the mixture is more significant as it shows a direct action on pancreatic β cells. | [43] |
| Clinical study | Considering the research of Van Proeyen et al. (63), the level of insulin stimulation by action of OFI combined with leucine (Leu) was compared. It was a randomized double-blind crossover study, where 11 subjects underwent an OGTT test after a cycling session. The study included an evaluation on whether this combination has an additive action on insulin stimulation after exercise. | After 60 min, the individuals ingested glucose and three types of capsules (some with 1000 mg OFI, others with a combination of OFI and Leu, and those with only Leu). Blood glucose and serum insulin were measured. The data showed that only the OFI group reduced blood glucose and the area under the glucose curve (AUGC). Furthermore, the OFI plus Leu group increased serum insulin concentration; suggesting that this combination may stimulate carbohydrate-induced insulin after doing exercise. | [44] |

**Table 3.** *Cont.*

| Type of Study | Objective and Characteristics | Results and Conclusion | Ref. |
|---|---|---|---|
| In vivo | Berraaouan et al., calculated the in vitro antioxidant potential of OFI seed oil (CPSO) and its protective effect against Allox-induced diabetes mellitus. | They used a DPPH-scavenging assay for the first objective. To evaluate the preventive effect, Swiss albino mice treated with CPSO (2 mL/kg, orally) were used before and after Allox administration. During the in vivo test, body weight and fasting blood glucose were measured and a histopathological analysis of the pancreas took place. CPSO showed a relevant antioxidant action; it also reduced hyperglycemia and protected the islets of langerhans against Allox-induced tissue changes. The conclusion is that CPSO decreases oxidative stress and inhibits lesions in pancreatic β cells. | [45] |
| In vitro | The objective was to analyze some ethanol fractions of cladode polysaccharides of *O. monacantha* (POMC). | POMC fractions IV and V were obtained by means of anion exchange chromatography and purified with a Sephadex G-50 gel filtration column. By gel permeation chromatography (GPC), high-performance liquid chromatography (HPLC) and gas chromatography (GC) it was established that POMC V had a molecular weight of 28.7 kDa and consisted mainly of rhamnose, arabinose and glucose; while POMC VI, had a smaller molecular weight (10.8 kDa) and was composed by rhamnose, mannose and glucose. | [46] |
| In vivo | The purpose of this study was to determine the most effective hypoglycemic component of OdHw polysaccharides and to study their antidiabetic ability in STZ-induced diabetic mice. | Initially, three types were identified (ODP-Ia, ODP-Ib, and ODP-II). After the administration of the ODP-Ia type, the food intake, blood glucose, and TG levels significantly decreased when measured in the fasting state. However, ODP-Ia did not increase insulin levels. It is suggested that ODP-Ia exerts its antihyperglycemic effect by protecting the liver and improving its sensitivity and cellular response. | [47] |
| In vivo | The aim of this research was to evaluate the effects of cactus pads extracts and pulp fruit on blood glucose concentration and the glycemic curve in Sprague-Dawley rats. | After 8 days of daily intake, the peaks and glycemic curves of the cactus pads and pulp fruit groups and the Cr (III) batch were less pronounced than those of the control group. In addition, a slight decrease in fasting blood glucose resulted. These data suggest that the Cr (III) content in these plant foods is related to their antihyperglycemic capacity. | [50] |
| In vivo | Ibañez and Meckes analyzed the hypoglycemic effect of a semi-purified fraction of OS in rabbits. | They confirmed that the powder fraction obtained produces an effect similar to the whole extract traditionally obtained from the stems of the vegetable. Their results suggest that the semi-purified product requires further evaluation to be considered a hypoglycemic agent. | [51] |
| In vivo | The antihyperglycemic effect of some edible plants (*Cucurbita ficifolia, Phaseolus vulgaris, OS, Spinacea oleracea, and Cucumis sativus*) was analyzed in healthy rabbits subjected to weekly tests of subcutaneous glucose tolerance. | Most of the plants had this capacity. However, OS was the one that most significantly decreased the area under the glucose tolerance curve and the hyperglycemic peak. | [52] |
| In vivo | Given that OdHw is traditionally used in the Canary Islands, the effect of its red juice on blood glucose levels in normoglycemic and Allox-induced diabetic rabbits was tested. | An oral dose of 5.0 mL/kg significantly reduced the increase in hyperglycemia in both types of rabbits. OdHw did not increase plasma insulin levels and was similar to that of an oral dose of tolbutamide (100 mg/kg). These data suggest that OdHw produces hypoglycemia mainly by reducing intestinal glucose absorption. | [53] |

**Table 3.** *Cont.*

| Type of Study | Objective and Characteristics | Results and Conclusion | Ref. |
|---|---|---|---|
| In vivo | The study was on hypoglycemic activity of an purified extract of *O. fuliginosa* (Of) in induced diabetic rats by STZ. | Blood glucose and glycated hemoglobin (HbA1c) levels were reduced to normal values by a combined treatment of insulin and extract. When the insulin was with-drawn from the combination treatment, the extract maintained the normoglycemic state in the diabetic rats. The mechanism of action induced by the dose of Of (1.0 mg/kg/day) is possibly related to its fiber content. | [54] |
| In vivo | The purpose of the study was to evaluate the nutritional value of OdHw and its curative potential in STZ-induced diabetic rats. | The results showed that OdHw is a rich source of fiber, carbohydrates, minerals and vitamins. In addition, oral administration of OdHw juice significantly reduced blood glucose levels and by means of a histopathological analysis of pancreatic tissue improvement in the cells of the islets of Langerhans was observed, which may explain the antidiabetic effect of OdHw. | [55] |
| In vivo | The effects of some extracts of *O. Milpa Alta* [Aqueous, petroleum ether, ethyl acetate (EtOAc), butanol (BuOH)] were tested in STZ-induced diabetic mice. | The results indicated that all the extracts managed to lower glucose levels; although petroleum ether extract was the most significant. | [56] |
| In vivo | This study focused on the effect of OFI on blood glucose metabolism of db/db mice treated for 4 weeks. | After this period, food intake, plasma glucose and insulin levels decreased markedly. Furthermore, a histopathological analysis showed that the morphology of the pancreatic islets improved in the animals treated with OFI. | [57] |
| Systematic review | The purpose of the investigation was to analyze the efficacy of some natural products (*Opuntia, Gymnema, Tecoma, Ginseng, Karela, Alpha lipoic* and *Panaxans*) commonly used for diabetes. | After a MEDLINE search of articles published between 1960 and 2001, nopal was found to be the most widely used herbal hypoglycemic agent in people of Mexican descent; while Karela is mainly used in Asian countries. Studies reveal different mechanisms of action, among which the high content of soluble fiber stands out. | [60] |
| In vivo | The hypoglycemic activity of an extract of *O. lindheimeri Englem* was investigated in STZ-induced diabetic pigs. | A dose-dependent decrease in blood glucose concentration resulted from the oral administration of two doses (250 and 500 mg/kg) of the extract. Furthermore, the greatest hypoglycemic effect appeared 4 h after the intake. The conclusion is that this experimental model can be useful to evaluate long-term effects of *Opuntia* consumption given the physiological similarities of pigs with humans. | [61] |
| Clinical study | A double-blind controlled study (obese prediabetic individuals of both genders) was performed on the acute and chronic effects of OFI. The OGTT test was evaluated with a bolus of 400 mg of OFI ingested 30 min before consuming glucose. | In the acute phase, a significant decrease in blood glucose concentrations was observed during the next 60, 90 and 120 min. On the contrary, in the chronic phase, no differences were observed with the evaluated schedules of the OGTT, in the blood chemistry variables (insulin, adiponectin, Hb1Ac) and in the body composition after 16 weeks of supplying 200 mg of OFI. | [62] |
| In vivo | Two extracts of *O. streptacantha* [cladode traditional extract (LE) and traditional filtered sample (FE)] were evaluated in diabetic rats with STZ by two tests. | The first was to confirm its hypoglycemic capacity (LE 135 mg/kg and FE 27 mg/kg) and the second was to quantify the antihyperglycemic potential using oral glucose tolerance test (OGTT). The conclusion was that both extracts did not produce a significant hypoglycemic effect but an antihyperglycemic action compared to a control group of animals. | [63] |

**Table 3.** *Cont.*

| Type of Study | Objective and Characteristics | Results and Conclusion | Ref. |
|---|---|---|---|
| Clinical study | Healthy men participated in a double-blind crossover study that included 2 experimental sessions. In the first one, they underwent OGTT at rest (OGTT-R) and cycling activity for 30 min. Immediately after the exercise, they received capsules containing 1000 mg of OFI extract and another OGTT (-EX) was performed. | Blood samples were collected at baseline and at 30-min intervals after the ingestion of 75 g of glucose in order to determine blood glucose and serum insulin. The results indicated that in OGTT-R, the AUGC was reduced by 26% and serum insulin had a higher concentration; while in OGTT-EX, glucose decreased approximately 10% lower with OFI compared to the placebo group. In conclusion, the extract can increase plasma insulin and facilitate the removal of an oral glucose load from the circulation at rest and after doing exercise. | [64] |
| In vitro In vivo | There is no information about whether the maturity stage in OFI can alter its antidiabetic capacity. Thus, the effect of small (SCF), medium (MCF) and large (LCF) cladode flours in diabetic rats was analyzed. | Only the MCF and SCF batches (50 mg/kg dose) showed a reduction in postprandial blood glucose. Furthermore, in vitro glucose diffusion tests showed a similar classification in both types of flour. It is considered was that the maturity stage alters the fiber content and produces differences in its viscosity, affecting in vitro and in vivo glucose responses. | [65] |
| | | Obesity | |
| Systematic review | A sedentary lifestyle and excessive calorie consumption are known to be key factors in the prevalence of obesity. In consequence, reducing dietary fat absorption through approved drugs and natural treatments could help control this health problem. | Information gathered from four randomized controlled clinical studies on the efficacy of Litramine IQP-G-002AS (fiber derived from OFI) in reducing fat absorption suggests that it is effective in promoting fat excretion and weight loss; especially when ingested at a daily dose of 3 g for seven days. | [58] |
| In vivo | The purpose of this research was to determine the metabolic effect of an OFI extract in a diet-induced obese mouse model. The extract was added to a high-fat diet and administered for 12 weeks. | The doses used (0.3 and 0.6%) prevented the C57BL/6 mice from presenting high values of LDL-Cho, HDL-Cho, tCho and increasing their body weight. An improvement in glucose tolerance and an increase in energy expenditure were registered. In addition, the extract stimulated insulin secretion in isolated pancreatic islets. The decrease in metabolic abnormalities was associated with a higher content of mRNA for glucose transporter 2 (GLUT2) and peroxisome proliferator-activated receptor gamma (PPARγ). | [59] |
| In vivo | To determine the nutritional potential of whole *O. ficus-indica* seeds (OFIws) and its effect on food intake, Wistar rats received a treatment based on a diet supplemented with OFIws for nine weeks in which the efficiency of feed conversion, the protein efficiency index, and body weight were observed. | The results indicated a significant decrease in blood glucose concentration and body weight; as well as an increase in HDL-Cho and glycogen in the liver and skeletal muscle. Which suggests that OFIws is a healthy and useful food for obesity treatments. | [66] |
| Systematic review | Today, hundreds of weight loss products are in the global dietary supplement market. However, their effectiveness has not been fully proven. Through an electronic search, the effectiveness of PPFs was analyzed using published data from randomized clinical trials. | Five studies which varied in the design and quality of the reports were included. The analysis revealed a significant reduction in body mass index, body fat percentage, and tCho. Adverse events included gastric intolerance and flu symptoms. It is recommended to increase the number of clinical trials to have more consistent data. | [67] |

Table 3. *Cont.*

| Type of Study | Objective and Characteristics | Results and Conclusion | Ref. |
|---|---|---|---|
| Systematic review | A bibliographic compilation was focused on the aspects of ethnobotany, toxicity, pharmacology, state of conservation, trade and chemistry of the medicinal plants used in Mexico, Central America and the Caribbean for the empirical treatment of obesity. | A total of 139 species were recorded, including *O. robusta* (OR), OM, OS, OM, OJ and OFI. The conclusions were: (a) There are no clinical studies in obese subjects using the medicinal plants mentioned in this review, (b) There are no herbal products approved in Mexico for the treatment of obesity, and (c) The need for other pharmacological, phytochemical, and toxicological studies with medicinal flora to obtain new antiobesity agents of high importance. | [68] |

*4.3. Hepatoprotective Effect*

The liver plays a fundamental role in the regulation of various physiological processes, and its activity is related to different vital functions, such as metabolism, secretion and storage. It has the ability to detoxify endogenous and/or exogenous substances, helps in the metabolism of carbohydrates and fats, in the secretion of bile and participates in the supply of nutrients and energy. Unfortunately, "liver disease" (a term that indicates damage to cells, organ structure or function) continues to be one of the main threats to public health around the world. This damage can be induced by biological factors (bacteria, viruses and parasites), autoimmune diseases (immune hepatitis, primary biliary cirrhosis), as well as by the action of different chemicals, such as some drugs [high doses of acetaminophen (APAP) and anti-tuberculosis drugs], toxic compounds [carbon tetrachloride ($CCl_4$), thioacetamide, dimethylnitrosamine (DMN), D-galactosamine/lipopolysaccharide (GalN/LPS)], mycotoxins (aflatoxin $B_1$) and undoubtedly excessive alcohol consumption [12].

Despite the various therapeutic uses attributed to the genus *Opuntia*, scientific research on its hepatoprotective capacity began in 2004, when Wiese et al [69] reported that OFI could reduce the symptoms (nausea, dry mouth and anorexia) characteristic of hangover after consuming excess alcohol. Subsequently, Galati et al. (2005) examined the effects of prickly pear fruit (JPPF) juice against $CCl_4$-induced hepatotoxicity.

After administering 3 mL of JPPF per rat, the liver parenchyma lesion was restored after 72 h. In addition, plasma levels of ALT and AST were reduced. The investigators suggested that hepatoprotection could be related to flavonoids, betalains, and vitamin C, that synergistically, act on the antioxidant activity of JPPF [70].

On the other hand, Ncibi et al [71] demonstrated that a cactus cladode extract (CCE) from OFI could reduce the liver toxicity of the (CPF). Such conclusion resulted when combining the pesticide plus CCE and achieving significant normalization of biochemical parameters: ALT, AST, alkaline phosphatase (ALP), lactate dehydrogenase (LDH), Cho and albumin (Alb), in contrast to animals treated with CPF only where the same parameters were notably affected. Research carried out by Dalel Brahmi's scientific group explored the protective potential of SCC against two hepatocarcinogenic agents in Balb/C mice: Benzo(a)pyrene [B(a)P] [72] and $AFB_1$ [73].

In both studies, the two carcinogens altered EOx markers, such as the level of malondialdehyde(MDA) and catalase activity (CAT), and increased the expression of heat shock proteins (Hsp 70 and Hsp 27). Likewise, the authors demonstrated that pre-treatment with SCC significantly decreased the oxidative damage induced in all the markers tested. In order to confirm the protective capacity of JPPF, in 2012, the team evaluated this property against liver injury induced by chronic consumption of ethanol (EtOH) in Wistar rats. Pretreatment of animals with EtOH plus JPPF (20 and 40 mL/kg body weight, orally) interestingly reduced biochemical markers of liver injury [such as ALT, AST, ALP, LDH, Cho, TG, and gamma glutamyl transferase (GGT)]. An improvement was also observed in lipid oxidation, glutathione (GSH) content, the activity of some antioxidant enzymes [such as SOD, CAT and glutathione peroxidase (GPx)], as well as histopathological lesions induced

by chronic ingestion of EtOH for 90 days. These results again suggest that this property could be attributed to its ability to end free radical chain reactions [74]. Considering that cyclophosphamide (CP) has a high toxicity associated with ROS overproduction, the ability to JPPF to reduce liver damage and cytotoxicity induced by this alkylating agent in mice was evaluated. Again, biochemical markers (AST, ALT, LDH, Alb) and EOx indicators [degree lipid peroxide (LPO) and MDA] were analyzed. Cytotoxicity was also studied by reducing nucleic acids, proteins, and glutathione in liver cells. After a pretreatment with JPPF, all mentioned liver markers were statistically restored [75].

Not only OFI has demonstrated a hepatoprotective potential, since González-Ponce et al. (2016) analyzed the antioxidant activity of OS and OR extracts against APAP-induced acute liver failure (ALF). They also administered both extracts (800 mg/kg/day, orally) to Wistar rats prior to APAP intoxication, which significantly attenuated lesion markers (AST, ALT, and ALP) and improved liver histology.

Furthermore, in a culture of hepatocytes, the extracts reduced LDH leakage and cell necrosis, both prophylactically and therapeutically. Apparently, OR showed higher levels of antioxidants than OS. These results suggested that both extracts could be considered as nutraceuticals to prevent ALF [76]. Finally and considering that OdHw appears to have antioxidant and anti-inflammatory properties, the hepatoprotective effect of a hydroalcoholic extract against lead acetate (Pb) -induced toxicity in an animal and cellular model was analyzed.

In the first case, Wistar rats received the extract (100 and 200 mg/kg/day) plus Pb for ten days. At the end of the period, it was observed that OdHw increased CAT activity and decreased the activity of MDA and serum liver enzymes (ALP, ALT, AST). In the case of the in vitro assay, HepG2 cells were used to measure three concentrations of OdHw (20, 40 and 80 µg/mL) on cell viability, MDA, GSH and TNF-$\alpha$. The measurement confirmed that all concentrations reduced the levels of MDA and TNF-$\alpha$ and increased at GSH [77].

As can be seen, most of the studies analyzed suggest that the main mechanism of hepatoprotective action is to act on the inhibition of EOx. However, because the antioxidant process remains largely unknown, activation of factor 2 related to nuclear erythroid factor (Nrf2) has recently been explored. In this sense, Nakahara et al. [78] demonstrated that an OFI extract showed potent antioxidant activity through the activation of Nrf2. Their conclusion was made by confirming that the antioxidant capacity of OFI was canceled in Nrf2 knockdown keratinocytes.

Cells where Nad (p) h: Quinone oxidoreductase 1 (NQO1) inhibits the generation of ROS induced with B(a)P and TNF-$\alpha$ have been evaluated. The results suggested that OFI upregulated Nrf2-NQO1 through activation of the aryl hydrocarbon receptor (AHR) and AHR-OFI binding regulated the expression of epidermal barrier proteins (such as filaggrin and loricrin) [78].

Recently, INS-1 cells were exposed to Allox with different concentrations of polysaccharides extracted from *O. Milpa Alta* (MAP) to measure the Nrf2 pathway and the activation of apoptosis in response to an increase in EOx. MAP restored cell viability and SOD and GSH activity, while considerably decreased the release of ROS, LDH, MDA and nitric oxide (NO) levels. Possibly, MAP may attenuate the apoptosis induced by Allox by increasing the expression of Bcl-2, decreasing the expression of Bax and the activities of caspase-3 and caspase-9 [79].

### 4.4. Effects on Human Infertility

Globally, statistics suggest that approximately 15% of couples show concern when they want to conceive and fail after 12 months of regular unprotected sex. For a long time, the female gender was wrongly stigmatized for its inability to conceive, but scientific advances in human reproduction of the last four decades have associated approximately 50% of this infertility with the male gender. Different environmental, physiological, and genetic factors have been identified in "male factor" infertility, especially those related to sperm dysfunction [80].

Oligozoospermia is a disease characterized by low sperm count and quality and responsible of 90% of male infertility. Unfortunately, routine semen studies and analysis have shown that not all men who have normal parameters are fertile. The hidden factor is now known to be EOx and is recognized as a major cause of idiopathic male infertility.

The fact that sperm contain a large amount of unsaturated fatty acids makes them prone to lipid peroxidation, causing DNA damage and activating its apoptotic elimination through a p53-dependent and independent mechanism that can lead to infertility. Some studies have shown that disorders such as poor fertilization, pregnancy loss, birth defects, and poor embryonic development are associated with sperm damaged by excessive EOx [80–83].

In a normal physiological state, seminal plasma contains an antioxidant enzymatic mechanism capable of quenching ROS and protecting sperm. However, a high level of ROS, triggered by factors such as inflammation, DM, AO, alcoholism, smoking, and environmental pollutants can minimize this protective mechanism [80,84–87].

Different studies have suggested that some food supplements such as selenium, zinc, carnitine and arginine increase sperm count and motility; while antioxidants such as vitamin E, C and B12, carotenoids, coenzyme Q and glutathione are beneficial for the treatment of male infertility. In other words, these compounds can help in the balance between ROS generation and the protective enzymatic mechanism of sperm [80,88,89].

In the specific case of *Opuntia* spp., there is little research that has explored its action in human infertility. Meama et al. (2012) conducted the first study and tested the effect of OFI on sperm DNA fragmentation (SDF) exerted by cryopreservation in two sperm populations [PI (brighter) and PI (dimmer)]. Normozoospermic men underwent semen analysis for infertility and their cell samples were subsequently cryopreserved in the presence of OFI extracts.

The process induced an increase in SDF only in the PI sperm population. In contrast, the addition of OFI slightly reduced SDF without affecting cell viability. Their results suggest that OFI probably prevents some damage to sperm during a cryopreservation process [90]. Some studies also suggest that antioxidant treatments administered in high doses can block the oxidative processes essential for the compaction of sperm chromatin.

Consequently, a trial was performed in couples with infertility and at least 2 attempts at assisted reproductive technology (ART) to evaluate a nutritional support (called Condensyl™) of the cycle of a pure carbon without strong antioxidants. The treatment consisted of a combination of B vitamins, zinc, *Opuntia fig* extract, small amounts of N-acetylcysteine, and vitamin E. A group of 84 patients consumed 2 tablets of Condensyl™ per day for 12 months.

The final results showed a positive response rate of 64% for the decondensation index and 71% for the DNA fragmentation index. Thus, 18 couples achieved a pregnancy before the planned ART cycle. The rest of the couples underwent a new ART attempt resulting in 22 additional pregnancies and 15 live births. The conclusion with these data was that low doses of Condensyl™ may have a positive potential on fertility by achieving a pregnancy rate and a live birth rate of 70% and 57%, respectively [91].

Another significant study was the one designed by Hfaiedh et al. (2014) to investigate the protective capacity of CCE (100 mg/kg) on sodium dichromate (SD) -induced testicular damage in male Wistar rats. After a 40 day treatment with SCC, it was possible to restore serum testosterone level, sperm count and motility to levels comparable to the DS control group. A reduction in the elevated level of lipid peroxidation and a significant increase in testicular SOD, CAT, and GPx activities were also registered [92].

Probably, the effect of CCE to minimize the oxidative damage induced by SD motivated the development of another in vivo test; the purpose of which was to analyze the reversible antifertility potential of two doses (300 and 900 mg/kg) of a methanolic extract from the fruit of *O. elatior Mill* (OeM). After administering OeM for 60 days, epididymal sperm count and motility were reduced by 80% without a decrease in serum testosterone levels.

On the other hand, testicular steroidogenesis or libido were not affected, unlike male fertility, which was suppressed when they were mated with virgin female rats. Said suppression of fertility was dose-dependent and reached the 100% at the highest dose. However, withdrawal of treatment for two weeks recovered sperm count, serum testosterone levels, and fertility [93].

Recently, Akacha et al. (2020) determined the role of an OFI ethanolic extract (EEOFI) in methotrexate (MTX) -induced testicular damage in rats. They considered using this chemotherapeutic agent due to its various drawbacks, especially in cells that are constantly dividing and developing. EEOFI (0.4 g/kg) was administered to rats treated with MTX and subsequently the sperm were collected and quantified where their motility was determined.

They also evaluated EOx markers (MDA, CAT, GPx and SOD) and marked serum testosterone levels by radioimmunoassay. The results confirmed that EEOFI had protective effects on rat gonad histology, oxidative stress, and sperm count and motility. Furthermore, serum testosterone levels increased considerably. Their results also suggest that EEOFI improves testicular injury and has a potent stimulating effects on fertility [94]. The contradictory results opens the field of research to confirm the antioxidant action of *Opuntia genus* on fertility.

### 4.5. Chemopreventive and Antigenotoxic Effects

The chemopreventive and antigenotoxic potential of *Opuntia* spp. is another field of research that has been widely explored by various scientists. The concept of cancer chemoprevention "Use of natural or synthetic biologically active substances that can prevent, inhibit, or reverse tumor progression" was established in the 1970s, and in the specific case of these opuntioid cacti, there have been conducted 21 studies (mainly in vitro and in vivo tests) to date. In general, the results suggest various mechanisms of action to try to prevent the development of this disease.

It is important to remember that the transformation of normal to malignant cells is driven by a multi-step process (conceptually divided into initiation, promotion, progression, invasion, and metastasis) due to genetic alterations that include mutations and/or epigenetic changes caused by genotoxic agents or genotoxins; which by their origin are divided into physical, chemical and biological.

Observations on cancer etiology reveal that as many as 90–95% of carcinoma cases are associated with chemical agents, only 5–10% with physical agents and around 2–5% with biological agents. However, considering that all mutagens are genotoxic, but not all genotoxic substances are mutagenic, the compounds that reduce DNA damage caused by genotoxins are called antigenotoxic and/or antimutagenic agents [3,95].

In general, antimutagens have been classified as desmutagens and bioantimutagens. The first group considers substances that promote the elimination of genotoxic agents from the body, as well as substances that partially or totally inactivate mutagens by enzymatic or chemical interaction before the mutagen attacks DNA.

On the other hand, bioantimutagens (known as true antimutagens) can suppress the mutation process after DNA is damaged and act on the repair and replication processes; resulting in a decrease in the frequency of mutations [3]. The reality is that the mechanisms of action of antigenotoxic and/or chemopreventive agents are varied and complex and it would be very difficult to fully explain them in this document. In summary, observations report that they act in different cellular and molecular events including apoptosis, cell proliferation, cell cycle, EOx regulation, DNA repair, activation/detoxification of carcinogens by xenobiotic metabolizing enzymes, functional inactivation/activation of oncogenes and tumor suppressor genes, angiogenesis and metastasis [3,95].

Practically, the clinical study (the only one so far) developed by Palevitch et al. (1993) where they treated benign prostatic hypertrophy (BPH) with a dried flower preparation of OFI was the one that started in this field of research. Their results showed that patients treated with 2 capsules (250 mg of dried flowers/capsule) orally, three times a day, for 6 and 8 months, significantly improved the discomfort associated with BPH. However,

they were unable to establish the mode of action of this preparation [96]. Table 4 shows the main studies carried out with different plant parts of *Opuntia* spp. that have demonstrated chemopreventive and antigenotoxic potential. In summary, OFI is the most studied species in in vitro models and in different types of extracts [hexane, EtOAc, acetone, methanol (MeOH) and aqueous ones]. Likewise, hexane extracts of its seeds, extracts (aqueous and EtOAc) of its PPFs, juices of its different varieties of PPFs [red-purple (PPRP), white-green (PPWG) and yellow-orange (PPYO)] and some of its bioactive compounds such as betanin (betacyanin isolated from its PPFs) and isorhamnetin glycosides. Another species explored is OHF (mainly, in hexane extracts, aqueous and EtOAc); and to a lesser extent, OR extracts, polysaccharides extracted from OdHw and *O. microdasys* at post flowering stage F3 (OMF3). Different cell lines have been used in these studies; the most representative are those extracted from the cervix, bladder, ovarian cancer (OVCA420), immortalized normal ovarian cells (SKOV3), human chronic myeloid leukemia (K562), breast cancer (MCF-7), colon cancer (HT -29), human glioblastoma (U87MG), lung squamous carcinoma (SK-MES-1), human BJ fibroblasts, and Caco-2, SW480, and HeLa cancer cells.

In the case of in vivo tests, only mice (Balb/C and NIH) have been used, causing them genotoxic damage by administering some mycotoxins [zearalenone (ZEN) and $AFB_1$] or a mutagenic agent [such as methyl methanesulfonate (MMS)]. Skin cancer has also been induced by 7,12-dimethyl-benz [a] anthracene (DMBA) and 12-*O*-tetradecanoylphorbol-13-acetate (TPA) and ultraviolet B (UVB) photocarcinogenesis.

In general, the frequency of micronuclei (MN), chromosomal aberrations in bone marrow cells and DNA fragmentation were quantified. Together, the results of these studies suggest that the chemopreventive and/or antigenotoxic effect of OFI, OHF, OR, OdHw and OMF3 is related to their ability to inhibit cell proliferation and induce apoptosis, accumulate ROS (pro-oxidant activity), anticlastogenic potential, modulate lipid peroxidation, induce phase II detoxifying enzyme system and antioxidant capacity. Among the bioactive compounds that exhibited these capacities are flavonoids (such as quercetin, kaempferol, isorhamnetin), betalains (such as betanin and indicaxanthin), carotenoids and phenolic compounds [4,73,97–114].

**Table 4.** Main studies of *Opuntia* spp. on its chemopreventive and antigenotoxic potential.

| Type of Study | Objective and Characteristics | Results and Conclusion | Ref. |
|---|---|---|---|
| In vitro In vivo | The purpose of the study was to evaluate the antioxidant capacity of three varieties of PPF juice [red-purple (PPRP), white-green (PPWG) and yellow-orange (PPYO)] in five different concentrations (100, 250, 500, 750 and 1000 mg/mL) by means of the DPPH method and selecting the variety with the highest antioxidant capacity to determine its anticlastogenic potential against MMS. | NIH mice were administered orally with PPRP and subsequently MMS was injected; which resulted in that PPRP was not a genotoxic agent, on the contrary, the reduction of MN frequency was proportional to the dose. | [4] |
| In vivo | The purpose of the study was to evaluate the antigenotoxic effect of CCE against $AFB_1$-induced damage in Balb/C mice. Animals were pretreated intraperitoneally with SCC (50 mg/kg body weight) for 2 weeks. | The results indicated that AFB1 induced significant alterations in EOx markers and was a genotoxic agent. In contrast, CCE reduced the number of chromosomal aberrations, DNA fragmentation, and the expression of p53 along with its associated genes (bax and bcl2). It is concluded that the genoprotective effect of CCE is probably related to its antioxidant capacity. | [73] |
| In vitro In vivo | In this study, the anticancer effect of five concentrations (0.5, 1.0, 5.0, 10 or 25%) of aqueous extracts of CLD from OFI in ovarian, cervix and bladder cells was evaluated; as well as in tumor growth in Balb/C mice. | After treating the cells for 3 and 5 days, an inhibition of cell growth and induction of apoptosis was confirmed in a dose-dependent and time-dependent manner. The extracts were also found to significantly suppress tumor growth and increase annexin IV expression in animals. | [97] |

**Table 4.** *Cont.*

| Type of Study | Objective and Characteristics | Results and Conclusion | Ref. |
|---|---|---|---|
| In vitro | The antiproliferative potential of betanin isolated from PPFs from OFI on the human chronic myeloid leukemia cell line (K562) was analyzed. | The results showed a decrease in the proliferation of K562 cells treated with a concentration of 40 μM. On the other hand, scanning electron microscopy revealed apoptotic characteristics such as chromatin condensation, cell contraction, and membrane blistering. While flow cytometry (FCM) showed 28% of cells in G0/G1 phase. In conclusion, betanin can induce apoptosis through the intrinsic pathway. | [98] |
| In vivo | Zourgui et al., analyzed whether EOx is a relevant parameter in the toxicity induced by ZEN and evaluated the efficacy, safety and antigenotoxic capacity of CCE to prevent the deleterious effects of ZEN. Balb/C mice were treated with the mycotoxin and three doses (25, 50 and 100 mg/kg b.w.) of CCE from OFI. | The results showed that ZEN increased the level of MDA, CAT and the generation of protein carbonyls in kidney and liver. While from the lowest dose of CCE the oxidative damage induced by ZEN was reduced. On the other hand, the same toxin induced MN frequency and chromosomal aberrations in bone marrow cells. This phenomenon was reversed by the three doses of CCE; emphasizing that the highest dose of the extract was safe and did not induce any genotoxic effect. These data suggest that SCC may reduce the detrimental effects of EOx and ZEN-induced genotoxicity. | [99,100] |
| In vitro | Nine PPFs juices from OFI were characterized in terms of color, pH, acidity, phenolic content, flavonoids, and betalains. The study included its antioxidant activity in vitro against four cancer cell lines [mammary (MCF-7), prostate (PC3), colon (Caco-2) and hepatic (HepG2)]. | In summary, the juices presented pH and acidity values that varied from 4.27 to 5.46 and from 0.03 to 0.27%, respectively. Variations were also observed in the content of flavonoids, betaxanthins and betacyanins. PC3 and Caco-2 were the cell lines most affected in their viability due to the action of PPF juices. | [101] |
| In vitro | Given that OHF has high concentrations of polyphenols and flavonoids, the anticancer effects of an EtOAc, aqueous and hexane extract on MCF-7 cells were investigated. | All extracts significantly decreased the number of viable cells in a concentration-dependent manner. Furthermore, a G1 arrest was induced in MCF-7 cells. In general, it was evidenced that the aqueous extract had a greater capacity to inhibit cell proliferation and induce apoptosis. | [102] |
| In vitro | Considering the previous studies of Yoon et al. (102) an extract of EtOAc, of hexane, and a fraction divided in water of *O. humifusa* (OHF) were again analyzed on cell proliferation, G1 arrest and apoptosis in U87MG human glioblastoma cells. | Cell proliferation was assessed using the MTT assay [3-(4,5-dimethylthiazol-2-yl) -2,5-diphenyltetrazolium bromide], and the effects of each extract on cell cycle and apoptosis were analyzed by FCM. The results were that both the hexane extract and the aqueous fraction reduced the number of viable cells. Furthermore, cell arrest was again induced in G1. | [103] |
| In vitro | Ovarian cancer cells (OVCA420) and immortalized normal ovarian cells (SKOV3) were treated with two concentrations (5 and 10%) of an aqueous extract of prickly pear (AEPP) from OFI. | After 2 days of treatment, both types of cells treated with AEPP showed a relevant increase in ROS. Specifically, high levels of DNA fragmentation and the expression of genes related to apoptosis (Bax, Bad, caspase 3, Bcl2, p53 and p21) that are sensitive to ROS were also observed in OVCA420 cells. After three days of treatment, the expression of NF-kappa B decreased, while p-AKT increased. The conclusion was that the inhibitory effect of AEPP on cell growth is through the accumulation of ROS and induction of apoptosis. | [104] |

**Table 4.** *Cont.*

| Type of Study | Objective and Characteristics | Results and Conclusion | Ref. |
|---|---|---|---|
| In vivo | Some in vitro tests have shown that OHF has anti-inflammatory, anti-proliferative and radical scavenging capabilities; Therefore, it was decided to evaluate its inhibitory effect on DMBA and TPA-induced skin cancer in Balb/C mice. | After previously feeding the animals with a diet containing 1.0 and 3.0% OHF, a reduction in the number of papillomas and epidermal hyperplasia occurred. The total antioxidant capacity, cutaneous glutathione S-transferase activity, and SOD also increased. Lipid peroxidation was measured in the skin cytosol and was only inhibited in the group fed 3% OHF. The results suggest that OHF exerts its chemoprevention by reducing EOx by modulating skin lipid peroxidation, enhancing antioxidant capacity, and inducing the phase II detoxifying enzyme system. | [105] |
| In vivo | The previous result of Lee et al. [105] that determined the chemopreventive capacity of OHF on skin cancer induced by DMBA and TPA motivated this new study to analyze the protective potential of OHF against UVB-induced photocarcinogenesis. Again, Balb/C mice were fed OHF and subsequently irradiated twice every week for 30 weeks. | The final evidence was that the diet inhibited UVB-induced epidermal hyperplasia, leukocyte infiltration, myeloperoxidase level, and pro-inflammatory cytokine levels. In addition, the presence of interleukin-1$\beta$ (IL-1$\beta$), IL-6, TNF-$\alpha$, the level of expression of mRNA and COX-2 were reduced. Taken together, these data suggest that such protection is associated with the inhibition not only of UVB-induced inflammatory responses involving COX-2 and pro-inflammatory cytokines, but also with the down-regulation of UVB-induced cellular proliferation. | [106] |
| In vitro | In this work, the residues from the juice production of PPFs derived from OFI and OR were explored as possible sources of natural chemotherapeutic ingredients against colon cancer. By means of a hydroalcoholic extraction and separation by adsorption, the natural extracts were produced and their antiproliferative effect was subsequently evaluated in the HT29 cell line (human colon carcinoma). | The results showed that the extracts inhibited cell growth and stopped the cell cycle, especially in G1, G2 and M. Betacyanins, ferulic acid and flavonoids (mainly isorhamnetin) are probably the main compounds responsible for cell cycle arrest. Besides, the death of cancer cells could have been induced by the pro-oxidant effect of these compounds. | [107] |
| In vitro | To analyze the antitumor effect of the polysaccharides extracted from OdHw on cells of squamous cell carcinoma of the lung (SK-MES-1), the AnnexinV assay, FCM and Western-blotting were used. | The results showed that different concentrations of polysaccharides inhibit the growth of SK-MES-1 cells and stop the cell cycle in phase S. The AnnexinV assay revealed the induction of apoptosis. These data suggest that cell inhibition and apoptosis may be attributed to an increased expression of the P53 protein and the tension homolog deleted on chromosome ten (PTEN) protein. | [108] |
| In vitro | The purpose of the research was to determine the antiproliferative effect of extracts of OFI and different isorhamnetin glycosides in two cancer cell lines (HT-29 and Caco-2). | The study showed that glycosides and extracts were more cytotoxic against HT-29 cells. A bioluminescent analysis revealed an increase in caspase 3/7 activity in cells treated with the extracts, while FCM confirmed that both extracts and glycosides induced greater apoptosis in HT-29 cells. However, isolated isorhamnetin was more apoptotic in the Caco-2 cell line. The conclusion was that glycosylation induces the antiproliferative effect exerted by isorhamnetin extracts and glycosides. | [109] |

**Table 4.** *Cont.*

| Type of Study | Objective and Characteristics | Results and Conclusion | Ref. |
|---|---|---|---|
| In vitro | The research was on the effect of an AEPP derived from OFI and its pigment Ind on the proliferation of Caco-2 cells. | Both compounds caused apoptosis in nutritionally relevant amounts and their action was dose-dependent. Despite this, Ind accounted for approximately 80% of the protective effect, although not inducing EOx in Caco-2 cells. Probably, the epigenomic activity of Ind was to demethylate the promoter of the tumor suppressor gene p16 and reactivate the expression of silenced mRNA, favoring cell inhibition in the G2/M phase. | [110] |
| In vitro | Initially, the bioactive compounds of different extracts (hexane, EtOAc, acetone, MeOH and MeOH: water) of cladodes from OFI were identified and quantified by HPLC. Subsequently, their chemopreventive activities were evaluated in two types of cells (MCF7 and SW480). | The results indicated that the acetone and MeOH extract showed the highest amount of polyphenolic compounds. Further to this, most of the extracts, with the exception of hexane, exhibited significant cytotoxicity in both cell lines; although the most sensitive was the SW480. These findings suggest that the cell death induced by the extracts caused an inhibition of cyclooxygenase-2 (COX-2) and increased the Bax/Bcl2 ratio, favoring apoptosis. The set of antioxidant, antiproliferative and proapoptotic activity of bioactive compounds probably promote their chemopreventive role. | [111] |
| In vitro | Considering that the total levels of polyphenol and ascorbic acid in OHF are high, the premise was that their antioxidant compounds could inhibit the survival of two cell lines [cervical carcinoma (HeLa) and human BJ fibroblasts]. | Hexane extracts from their seeds and EtOAc extracts from PPFs and CLD significantly suppressed HeLa proliferation, but did not affect BJ fibroblasts. Another observation was that G1 phase arrest was induced in HeLa cells, which was associated with low levels of cyclin D1 [cyclin-dependent kinase 4 (Cdk4)]. This result motivated to examine the EtOAc extract on the tumor growth of the HeLa cell xenograft, due to the finding that the tumor volume had been reduced; which was correlated with the decrease in the expression of Cdk4 and cyclin D1. It is suggested that both extracts may be promising candidates for the treatment of human cervical carcinoma. | [112] |
| In vitro | This study involved two objectives; the first, to evaluate the analgesic and anti-inflammatory activity of OMF3; and the second, to determine its antigenotoxic effects in Allium cepa test. | By means of the acetic acid contortion test and the carrageenan-induced foot edema test, OMF3 showed to have high analgesic and anti-inflammatory activity (72 and 70%, respectively). OMF3 also induced an antimutagenic potential at a concentration of 60 μg/mL against H2O2-induced damage. | [113] |
| In vitro | Despite the fact that OFI is an important dietary source and a traditionally used medicinal plant, there are few studies on its toxic effects. Therefore, a toxicological evaluation was carried out using the 3-(4,5-dimethyl-2-thiazolyl)-2,5-diphenyl-2H-tetrazo-lium bromide (MTT), Comet and the γH2AX In-Cell Western Assay. | None of the extracts showed any cytotoxic or genotoxic effect on the HepG2 cell line; on the contrary, both the fruit pulp and the extracts of seeds, flowers and cladodes showed a protective effect against the genotoxicity induced by H2O2. This evidence suggests that OFI extracts do not have cytotoxic and/or genotoxic effects. | [114] |

## 5. Conclusions and Perspectives

The investigations shown in this review demonstrate the nutritional, medicinal, pharmaceutical and preventive impact of the different species of *Opuntia* spp. However, they also reveal the possibility of expanding and conducting new studies (in vitro, in vivo and clinical) in order to confirm its different mechanisms of action that together favor its beneficial properties. In general, the antiatherogenic, antihyperlipidemic, antihypercholes-

terolemic and antidiabetic effect share two mechanisms of action; the first one related to the soluble fiber content that decreases body weight and slows down the speed of digestion and/or intestinal absorption of glucose and fatty acids. The second, undoubtedly, lies in its antioxidant property, which is directly related to the role that OXs plays in the development of atherosclerosis and cardiovascular diseases that are complications of DM2 and OA. This mechanism is related to the presence of some flavonoids, phenolic compounds and fatty acids, specifically quercetin 3-methyl ether and/or omega-6 linoleic acid from cactus seed oil, both with hypocholesterolemic effects. Likewise, betalains (such as indicaxanthin and betanin) protect the vascular endothelium from inflammation and cytokine-induced oxidative alteration, such as TNF-$\alpha$. Also, the polysaccharide content has shown anti-inflammatory activity and the ability to reduce lipoperoxidation and/or sequester ROS. In the control of DM2, the hypothesis that the ingestion of nopal improves the postprandial glucose response and stimulates insulin secretion through a direct action on $\beta$ pancreatic cells has also emerged.

In addition, the proposal that Cr (III), present in CLD, balances carbohydrate and lipid metabolism, favors a positive effect on insulin signaling and/or function, improves its systemic sensitization and reduces plasma glucose. After studies related to ASCVD, DM2 and OA, another widely explored field of research is its chemopreventive and antigenotoxic potential; where approximately 21 studies (mainly in vitro and in vivo tests) have suggested various mechanisms of action, highlighting the induction of apoptosis, inhibition of cell proliferation and cell cycle, activation of the phase II detoxifying enzyme system and DNA repair.

Furthermore, the antioxidant and/or regulatory effect of EOx has been included. These actions have also been considered as the main hepatoprotective mechanism of *Opuntia* spp. The antioxidant process has been analyzed for many years; yet, it remains largely unknown. So it would be interesting to increase research exploring the activation of Nrf2. In the case of the action of *Opuntia* spp. on human fertility, there is little research and the results are contradictory since some studies suggest that antioxidant treatments administered in high doses can block the oxidative processes essential for the compaction of sperm chromatin; which opens another field of investigation. Taken together, all the studies point to the conclusion that the CLD and PPFs from *Opuntia ficus-indica* (OFI) are the plant parts and the species that have been studied the most. Regarding the form of analysis, hexane, EtOAc, acetone, MeOH and aqueous extracts have been evaluated. Besides OFI, other species studied are OdHw, OHF, OS, OH, OA, OM and OR. It is convenient to remember that the Cactaceae family contains approximately 130 genera and 1500 species, which favors a wide genetic diversity that in conjunction with environmental conditions (climate, humidity), soil type, age of maturity of the cladodes and the harvest season generates differences in the phytochemical composition of their plant parts (PPFs, CLD, roots, flowers, seeds and stems) between wild and domesticated specie, inducing changes in its nutritional values and indisputably in its functional and/or therapeutic properties. Furthermore, although the public and some health care professionals believe that herbal medicines are relatively safe because they are "natural", there are remarkably little data to support this assumption. Therefore, *Opuntia* spp. species are not exempt from possible adverse and toxic effects. In general, OFI has been found to be well tolerated orally, even at high doses, presenting on certain occasions mild diarrhea, increased volume and frequency of stool, nausea, headache and low colonic obstruction [14,114,115]. Both aspects (genetic diversity and toxicology of *Opuntia* spp.) will be analyzed in more detail in part 2 of the manuscript.

In conclusion, by combining all the information in this review favors the field of research in the biotechnology area where new studies could be developed with other species to explore their capacities and pharmacological properties, their doses and administration intervals and analyze their possible toxic effects in the medium and long term. Likewise, the bioactive compounds extracted from the different species could be used in the preparation of drugs and nutraceuticals, as well as to obtain chemopreventive agents directed at cancer and/or chronic degenerative diseases. Future studies on different pure opuntiode

cacti, extracts and isolated bioactive compounds will allow a greater understanding of the properties of *Opuntia* spp., which is a genus of plant consumed by humans for more than 8000 years, with a high frequency throughout the world and which is apparently considered a safe plant.

**Author Contributions:** E.M.-S., E.M.-B., J.A.M.-G., J.P.-R., N.V.-M. designed the concept, wrote the majority of the paper and managed the authors; P.E.M.-G., I.Á.-G., J.A.I.-V., M.S.-G., L.D.-O. conducted the literature search, wrote key sections of the paper; Á.M.-G., L.A.-R. and T.F.-A. wrote sections of the paper and managed the reference list. All authors have read and agreed to the published version of the manuscript.

**Funding:** This research received no external funding.

**Institutional Review Board Statement:** Not applicable.

**Informed Consent Statement:** Not applicable.

**Data Availability Statement:** Not applicable.

**Acknowledgments:** The authors thank Florencia Ana María Talavera Silva for all her academic support. Her comments and observations in reviewing articles are always valuable and we give her immense recognition for her efforts.

**Conflicts of Interest:** The authors declare no conflict of interest.

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
