# Peer review of "Opuntia genus in Human Health: A Comprehensive Summary on Its Pharmacological, Therapeutic and Preventive Properties. Part 1"

_horticulturae, doi:10.3390/horticulturae8020088_

Round 1

Reviewer 1 Report

Dear Authors,  

In the article entitled » Opuntia genus in Human Health: A Comprehensive Summary on Its Pharmacological, Therapeutic and Preventive Properties. Part 1” the extensive review about scientific evidence of the Opuntia genus beneficial properties on human health was done. The proposed health effects were described clearly and in detail. Many publications were included.

Although the results presented are important and should be published, my main concern is to better explain and involve also studies addressing the adverse effects of the application of the proposed plant extract in high amounts. In addition, and in more detail, other concerns are: 

Line 32: “action” can be replaced with “preventive effects against”

Line 135: The variability of the compounds found in the Opuntia genus should be described. Especially the factors that affects the variably should be addressed such as genetic and environmental factors.

Table 1:  References should be included in the table, next to the values. Why some concentrations of the compounds are missing?

Line 526: Describe the compounds having these properties.

Line 538:  Include the chapter or a few sentences about possible adverse effects. What can happen if the plant extract is applied at higher doses. What are these high values?

Line 549: The polysaccharides are mentioned as the main compounds having beneficial properties in Opuntia genus. What about other compounds?

Substantial changes should be carried out before acceptance.

Good luck!

Author Response

Answers

Dear reviewer

The authors appreciate the comments and observations of the article

We have considered all suggestions and observations

Please check the attached file

Thanks for everything

Receive a cordial greeting

Line 32: We include the phrase "preventive effects against”

Line 135: We include a small paragraph (On line 135 and in the conclusions/perspectives section) where the importance of genetic variability and environmental conditions that generate differences in the phytochemical composition between wild and domesticated species and induce changes in their nutritional and functional properties

Table 1: We have included the data on the concentrations of the missing compounds. We have also included the reference of the articles from which the most significant data were obtained.

Line 526: We have included a short paragraph that describes the compounds with these properties.

Line 538: We have included a small paragraph on the possible adverse effects of Opuntia spp.

We do not think it is appropriate to attach a larger section on the genetic diversity and toxicology of Opuntia spp since they will be analyzed in more detail in part 2 of this manuscript.

Line 549: In addition to the polysaccharides that we already mentioned. We have also included a small paragraph that mentions the beneficial effect of other bioactive compounds.

Reviewer 2 Report

The manuscript presents a valuable but not sufficiently novel approach to the preventive and therapeutic pharmacological properties of the genus Opuntia. A significant number of scientific information has been generated in this regard, so the authors must improve the focus and depth of information analyzed.  Especially the authors have to think if it is worth generating a second review manuscript on the subject. In this sense, I suggest building a more solid document that can contemplate other approaches for discussion based on the scientific literature on these issues.

The authors should try to modify the focus of the information presented in the different tables. The presentation is not very pleasant for the reader,  there is a lack of structure and order in the information included. I suggest summarizing and grouping them in a better way, for example by not showing and discussing individual studies, but rather grouping them by elements of common interest. In this sense, the objective of the studies could be separated from the conclusions, favoring a better understanding by the reader.

I suggest including some pictures that show the principal characteristic of the representative species Opuntia genus. Besides is very important to improve the discussion about the relationship between antioxidant activity and the biological mechanisms presented through the manuscript.

Author Response

Title: Opuntia genus in Human Health: A Comprehensive Summary on Its Pharmacological, Therapeutic and Preventive Properties. Part 1

Manuscript ID:  Horticulturae-1491770

Reviewer 2

Review Report (Round 1)

Comments and Suggestions for Authors

-To improve the information consigned in the different tables authors could be to change their structure. I suggest that the information be better grouped, not by years but by related studies and trials.

-In the same way, it is important that the information contained in the tables be better condensed. I suggest summarizing the information and pointing out only the most relevant aspects of each study.

-On the other hand, for a better discussion of the information, the authors can focus their attention on more related bioactivities, for example, action in atherosclerotic cardiovascular diseases, diabetes, and obesity could be analyzed in a more complete way

Answers

Dear reviewer

The authors appreciate the comments and observations of the article

We have considered all suggestions and observations

Please check the attached file

Thanks for everything

Receive a cordial greeting

We have changed the structure of the tables (The information was grouped by study and trials). The information was also reduced in some paragraphs (the most relevant aspect of each study is indicated).

At the end of the manuscript, a section with conclusions and perspectives (not necessarily discussion) was showed. Therefore, we try to be concrete and concise.

However, we have considered the suggestion.

We have included in this last section the most relevant conclusions from the information contained in the manuscript.

The conclusions were organized by preventive effect (atherosclerotic cardiovascular diseases, diabetes and obesity, hepatoprotection, human infertility and chemopreventive and/or antigenotoxic capacity).

In addition, we have included some small paragraphs about other information

Round 2

Reviewer 1 Report

Dear Authors,

In my opinion the manuscript could be now acceptable for publication.

Good luck!

Reviewer 2 Report

After reviewing the corrected versión I could verify the improvement of the manuscript. Best wishes whit the next editorial steps.